# A Rare Case Report of Metastatic Urothelial Carcinoma to Skull with Significant Reossification after Pembrolizumab

**DOI:** 10.3390/medicina57090987

**Published:** 2021-09-18

**Authors:** Yung-Hao Liu, Meng-Han Chou, En Meng, Chien-Chang Kao

**Affiliations:** Division of Urology, Department of Surgery, Tri-Service General Hospital, National Defense Medical Center, No. 325, Section 2, Cheng-Gung Road, Nei-Hu, Taipei 114, Taiwan; michael40315@yahoo.com.tw (Y.-H.L.); princecharmingben@gmail.com (M.-H.C.); en.meng@gmail.com (E.M.)

**Keywords:** metastatic urothelial carcinoma, skull metastasis, pembrolizumab, reossification

## Abstract

*Background*: Urothelial carcinoma ranks as the fourth most common cancer in men in the U.S; upon diagnosis, 10–15% have metastasized, mostly to lymph nodes, liver, lung, bone, and adrenal glands. Very few cases of skull invasion have been reported, and there is no established definite treatment. *Case presentation*: A 64-year-old Taiwanese male presented with metastatic urothelial carcinoma (mUC) of bladder with skull invasion. A sunken forehead without painful sensation could be palpated. After failure of chemotherapy, the patient received immunotherapy pembrolizumab, and complete remission of distant metastasis with reossification of osteolytic skull were noted. *Conclusion*: Immunotherapy has been reported to show significant remission in mUC, but mostly in solid organs or bone. While skull metastasis usually suggests late progression of the disease, immunotherapy has fewer systemic adverse effects than chemotherapy, and should be taken into consideration as a first-line therapy.

## 1. Introduction

The most common symptom in bladder cancer is painless hematuria, and it is usually diagnosed by urinary cytology and transurethral tumor resection. More than 90% of bladder cancers are urothelial carcinoma, 5% are squamous cell carcinoma, and less than 2% are adenocarcinoma [1]. Upon diagnosis, 35% are still organ confined while 10–15% have metastasized [2]. The National Comprehensive Cancer Network (NCCN) guidelines suggest that metastatic urothelial carcinoma (mUC) be treated with chemotherapy or concurrent radiotherapy; in cases of subsequent progression or recurrence, immunotherapy can be an alternative with fewer adverse effects and better survival [3]. However, among all the common metastatic sites, skull is rarely reported, and no definite treatment is suggested yet. We report a patient with mUC to skull showing significant remission of the disease and reossification of destructed skull lesions after pembrolizumab treatment.

## 2. Case Presentation

A 64-year-old Taiwanese male, with a history of benign prostate hyperplasia, presented with painless gross hematuria for a week. Cystoscopy disclosed a tumor over his right posterior bladder wall, followed by transurethral resection. The pathology report revealed high-grade infiltrating papillary urothelial carcinoma of the bladder. Pelvic magnetic resonance imaging (MRI) showed lymph node involvement. Chest computed tomography (CT) revealed metastatic nodules over the lungs and mediastinum. A whole-body bone scan revealed intensely increased uptake over the upper frontal skull, left scapula, left fifth rib, right femoral head, and right distal femur, suggesting multiple bony metastases. On physical examination, sunken forehead without pain or neurologic symptoms was palpated. For further detail, cranial CT revealed osteolytic bony destruction over the frontal bone and a homogenous tumor in the right retrobulbar orbital cavity. (Figure 1). He was diagnosed with UC of the bladder, cT3N3M1, stage IV, with multiple metastases.

After 21 cycles of cisplatin-based chemotherapy with altered regimens for one year, the disease progressed (Figure 2) and the patient underwent suffering with poor life quality, including cachexia, and required long-term bed rest. He was therefore shifted to immunotherapy pembrolizumab. After seven courses of pembrolizumab (one course = 200 mg triweekly), follow-up positron emission tomography (PET) showed prominent remission of distant metastases with no more abnormal FDG uptake throughout whole body, and cranial CT disclosed a reossified frontal skull (Figure 3). Even though imaging suggested skull lesion recovery, the sunken forehead due to bony destruction was still palpable without discomfort, but according to the patient, the depth of depression had slightly improved compared with previous clinical symptoms (Figure 4). During the immunotherapy treatment course, the patient presented with minor skin rash over bilateral upper limbs, but this subsided after skin lotion and occasional use of anti-histamines. The patient is currently still under treatment with an Eastern Cooperative Oncology Group (ECOG) performance status score of 0–1, suggesting that he was fully ambulatory and capable of self-care. Follow-up annual PET scans showed no significant FDG uptake in the following 2 years.

## 3. Discussion

In mUC patients, statistics from Babaian et al. showed that the most common metastatic sites are lymph nodes, liver, lung, bone, and adrenal glands. Of 107 cases in the study, only one was found to have metastasis to skull [4]. Another case of skull metastasis was reported by Chan FH et al. [5].

To treat mUC, after initial thorough workup to evaluate the extent of the disease, platinum-based chemotherapy remains the standard treatment [6]. Pembrolizumab, a humanized monoclonal IgG4κ isotype antibody against programmed death 1, has become a second-line therapy for patients with advanced or metastatic UC after platinum-based regimen failure [7]. In the phase III Keynote-045 trial, cases of progressed urothelial carcinoma after chemotherapy were recruited and randomly assigned to pembrolizumab and chemotherapy groups. Median overall survival was found to be longer in the pembrolizumab group (10.3 months) than the chemotherapy group (7.4 months) [7]. However, in the trial, except for the exclusion of liver and unstable brain metastases, metastatic sites of the recruited patients were not depicted.

In our case, the patient was found not only with metastasis to bones, but also to skull, which is, to our knowledge, the third reported case. The previous case presented with a painless swelling in the left occipital region [5]. X ray showed an osteolytic defect of the occipital bone. She underwent surgical resection of the tumor and radiotherapy. Pathology of the occipital tumor later revealed it as mUC. The patient died 3 months after the surgery due to gastrointestinal bleeding and uremia. Although the death may not have been related to her mUC, considering her age and status, pembrolizumab could have been administered prior to surgery or chemotherapy, as it causes fewer systemic adverse effects. However, since it is not easy to obtain tissue from skull lesions, one of the limitations may be the lack of pathohistology proof if we were to administer pembrolizumab on a suspected mUC patient.

So far, there have been no studies elaborating the treatment of “skull” metastasis specifically, but for patients with mUC to bone, which is a significant predictor of worse outcomes [8], it is suggested they receive chemotherapy or focal palliative radiotherapy, and if tumors express PD-L1 or if patients are not eligible for platinum-containing chemotherapy, then immunotherapy should be applied. Since the patients found with skull invasion are usually at a late stage of disease progression, they may not be candidates for surgery, and chemotherapy may aggravate their already poor physical status. A case of complete reossification of mUC to humerus after immunotherapy was reported [9]. His disease had progressed while on cisplatin-based chemotherapy. From this experience and ours, it is fair to say that surgery or chemotherapy for bone or skull lesion may not be the first option; instead, with fewer systemic adverse effects, immunotherapy should be taken into consideration as the alternative therapy in bone or skull metastatic patients. Our case demonstrates not only the rarity of mUC to skull, but also the fact that it is still treatable without compromising the patients’ life quality.

## 4. Conclusions

Metastatic UC to skull indicates a late stage of disease progression and a worse outcome. Current guidelines suggest chemotherapy or palliative radiotherapy. We herein report a rare case of mUC to skull, showing complete reossification of osteolytic skull lesion after immunotherapy pembrolizumab, without compromising his ECOG performance status. Immunotherapy should be considered early if there is a poor response to chemotherapy or surgical removal of tumor.

## Figures and Tables

**Figure 1 medicina-57-00987-f001:**
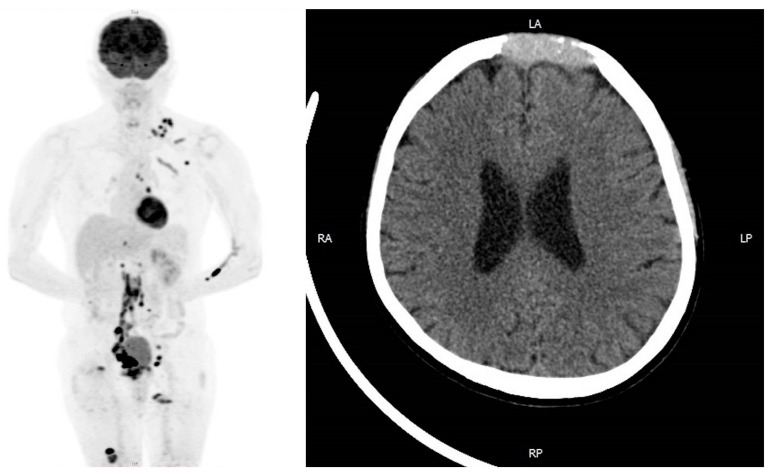
PET and cranial CT before treatment. The patient was diagnosed with urothelial carcinoma of the bladder, cT3N3M1. PET scan showed multiple metastases all over the body. Cranial CT revealed osteolytic skull metastatic lesions over frontal bone.

**Figure 2 medicina-57-00987-f002:**
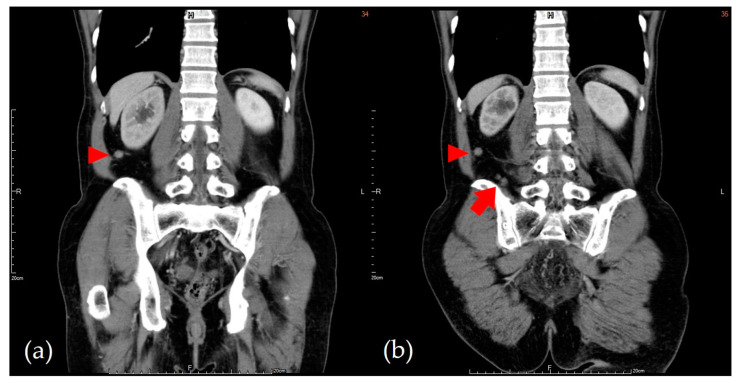
During the chemotherapy treatment course, abdominal CT was used to follow up the patient’s disease. Comparing images from before chemotherapy (**a**) and peri-chemotherapy (**b**), several new retroperitoneal nodal lesions (arrow) could be noted. Lymph node metastatic progression was favored.

**Figure 3 medicina-57-00987-f003:**
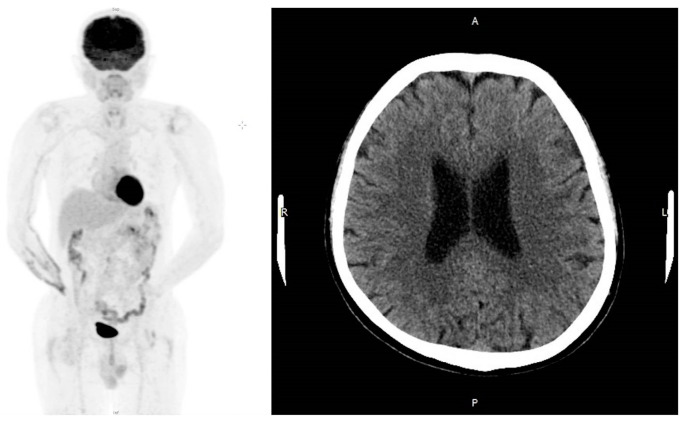
PET and cranial CT after pembrolizumab treatment. After treatment with unsuccessful chemotherapy followed by switching to immunotherapy pembrolizumab, PET scan disclosed complete resolution and brain CT revealed reossification.

**Figure 4 medicina-57-00987-f004:**
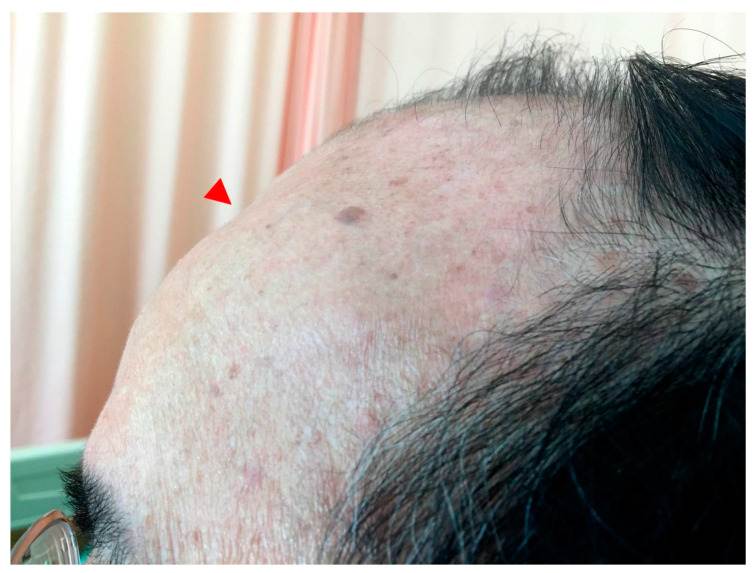
Physical examination of the forehead after pembrolizumab treatment. As the brain CT showed reossification, physical examination of the patient’s forehead still disclosed sunken with soft-tissue texture (arrowhead). No painful sensation was complained of.

## Data Availability

The data presented in this study are available on request from the corresponding author.

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
