# Peer review of "A Rare Case Report of Metastatic Urothelial Carcinoma to Skull with Significant Reossification after Pembrolizumab"

_medicina, 2021, doi:10.3390/medicina57090987_

Round 1

Reviewer 1 Report

It is very interesting to see such dramatic response to a metastatic site after initiation of Pembrolizumab, however, the real clinical impact is unclear.

Author Response

Dear reviewer,

As non-native English speaker, in order to improve readability, we have enclosed a certificate of English correction.

Best regards,
Yung-Hao Liu

Reviewer 2 Report

Authors reported a rare case of metastatic Urothelial Carcinoma to Skull with significant reossification after Pembrolizumab. Few questions need to be addressed 

  • Can authors show a biopsy from the patient with the expression of PDL-1 on tumor cells
  • During Immunotherapy did the patient present any systemic adverse effects and if yes please mention it in the manuscript.
  • Authors showed a PET and CT at two time point before treatment (Chemotherapy) and after treatment (Immunotherapy) however its lakes a PET and CT just before the treatment with Pembrolizumab was any performed before immunotherapy? To add it to the manuscript.
  • In the discussion authors argue the use of Immunotherapy as first line treatment in advanced stage based on there study and a previous study showing a remission after immunotherapy. However, in both case we cannot roll out the effect of chemotherapy because both patients were treated with chemotherapy before starting immunotherapy. Chemotherapy may have a role in killing cancer cells and release of tumor antigens that can activate T cells rendering Pembrolizumab more effective in the presence of activated T cells. Hence it may also interesting to discuss this point and study the possibility of combination therapy chemo + immunotherapy in patient with advanced stage.

Author Response

Dear reviewer,

Please see the attachment. Thank you very much for your time!

Best regards,
Yung-Hao Liu

Reviewer 3 Report

The author presented a really rare case in this case report entitled “A Rare Case Report of Metastatic Urothelial Carcinoma to Skull 2 with Significant Reossification after Pembrolizumab”.

This report is no doubt a unique case that Liu et al., pointed out. It is nicely written and concisely drafted.

I have few minor comments those will improve the quality of the manuscript.

Line 28: what NCCN stand for?

Line 61: Doses of Pembrolizumab will benefit the reader

Overall, comments, is the re-ossification due to long-term chemo followed by immunotherapy? Or it is the side effect of immunotherapy?

Author Response

(The authors gave the same response as above.)

Round 2

Reviewer 2 Report

Authors made the changes required for publication.